# Herbal Medicine for Postpartum Pain: A Systematic Review of Puerperal Wind Syndrome (Sanhupung)

**DOI:** 10.3390/healthcare11202743

**Published:** 2023-10-16

**Authors:** Na-Yoen Kwon, Hee-Yoon Lee, Su-In Hwang, Soo-Hyun Sung, Su-Jin Cho, Young-Jin Yoon, Jang-Kyung Park

**Affiliations:** 1Department of Obstetrics and Gynecology, College of Korean Medicine, Ga-Chon University, Seongnam-si 13120, Republic of Korea; kwonnay@gachon.ac.kr; 2Department of Korean Medicine Obstetrics and Gynecology, Pusan National University Korean Medicine Hospital, Yangsan-si 50612, Republic of Korea; leehy2722@pusan.ac.kr (H.-Y.L.); yyj@pusan.ac.kr (Y.-J.Y.); 3Department of Korean Medicine Obstetrics and Gynecology, School of Korean Medicine, Pusan National University, Yangsan-si 50612, Republic of Korea; hwangsi1216@gmail.com; 4Department of Policy Development, National Institute of Korean Medicine Development, Seoul 04554, Republic of Korea; koyote10010@nikom.or.kr; 5Research Institute of Nursing Science, Pusan National University, Yangsan-si 50612, Republic of Korea; chosujin13@gmail.com

**Keywords:** puerperal wind syndrome, herbal medicine, systematic review, meta-analysis

## Abstract

Mothers in the postpartum period often experience musculoskeletal disorders and pain, impacting their ability to care for themselves and their infants. Conventional treatments have limitations, prompting interest in alternative options like herbal medicine. This systematic review aimed to confirm the effectiveness and safety of herbal medicine treatment to improve maternal health in patients with postpartum pain (puerperal wind syndrome). We searched eight electronic databases for randomized controlled trials (RCTs) to evaluate the effects of herbal medicines on puerperal wind syndrome. Nine RCTs, including 652 patients, were selected. Following a meta-analysis of RCTs, both herbal medicine and combination treatments improved the visual analog scale scores, total effective rate, scores of Traditional Chinese Medicine syndromes, Oswestry Disability Index, and quality of life in patients with role-emotional puerperal wind syndrome. All adverse events were minor, and the incidence rate was not high compared with that of the control group. In conclusion, herbal medicine supports the improvement in pain, other systemic symptoms, and the quality of life of patients with puerperal wind syndrome. Moreover, no serious side effects were observed; therefore, herbal medicines appear to be safe. It can be the preferred treatment option for puerperal wind syndrome, which is currently managed symptomatically.

## 1. Introduction

After delivery, mothers undergo rapid physical and mental changes and are given the new social role of motherhood; therefore, they are placed in a vulnerable situation [1].

Puerperal wind syndrome (Sanhupung) was previously regarded as a cultural disease in a country, but now various studies [2,3] on postpartum pathological conditions are being conducted in recognition of the need for postpartum care.

Only when puerperal wind syndrome is not categorized as a disease and is confirmed to have the temporal incidence characteristics that followed labor and miscarriage can it be identified [4]. Degenerative disease, rheumatic illnesses, and intervertebral disc hernia are disorders that should be distinguished from puerperal wind syndrome in terms of musculoskeletal pain [5,6]. Thyroid disease, diabetes, and pituitary dysfunction must be separated from other illnesses when discussing fatigue, a cold, and excessive sweating [7]. Imaging and laboratory tests can be used to distinguish between these disorders [8]. However, even if the condition is identified, puerperal wind syndrome should not be quickly ruled out, as these illnesses can also occur with puerperal wind syndrome.

Many postpartum women develop musculoskeletal disorders due to continued hormonal influences, improper breastfeeding positions, and the biomechanical and ergonomic stresses of childcare-related activities [9,10]. To optimize maternal health, postpartum care needs to be provided continuously according to the mother’s needs. Body pain is one of the factors to be managed during the postpartum period [11].

The bodily pain experienced by mothers reduces their ability to take care of themselves and their quality of life (QOL) and can lead to postpartum depression. If the pain progresses chronically, it can cause functional disability and reduced work ability for the rest of their lives [11,12,13]. In addition, postpartum pain adversely affects baby care, causing negative maternal–infant attachment that is not controlled by mood, and secondary depression can adversely affect infant neurocognitive development [13,14]. Therefore, it is necessary to observe bodily pain during the postpartum period and intervene early to improve short- and long-term postpartum outcomes [13].

The conventional postpartum body pain is managed by symptomatic treatment with non-steroidal anti-inflammatory drugs (NSAIDs), opioids, and local anesthetics; however, these are limited during breastfeeding, and side effects, such as gastrointestinal upset, pruritus, constipation, and allergic or anaphylactic reactions, may occur [10,15,16,17]. Therefore, there is an increasing interest in alternative medical care for the treatment of postpartum body pain. In North Carolina, the alternative medicinal treatments most frequently used by midwives after childbirth were herbal medicines, massage, Chuna, and acupuncture [18].

According to Traditional Chinese Medicine (TCM), pain is caused by obstructions or deficiency of the Qi (vital energy) or blood and external pathogenic factors, such as wind, cold, dampness, and heat. For the treatment of pain, achieving holistic body balance by reinforcing a healthy Qi and eliminating pathogenic factors is important [15,19,20]. Traditionally in East Asia, including Korea and China, postpartum care is widely used to restore the weakened mind and body to a healthy state before pregnancy [21], and if the postpartum care is not performed properly, “puerperal wind syndrome (Sanhupung)” may occur [6]. Its symptoms are mainly pain and may include numbness, heaviness, or coldness of the body that occur after childbirth or miscarriage; there is no obvious abnormality on imaging or blood tests, and it is not classified as a disease. However, physical discomfort can adversely affect the mother’s QOL, mental health, and childrearing [6,22,23].

Herbal medicine, which refers to the utilization of plants or plant-derived materials, has long been used in many countries to treat pain [17] and various female diseases, such as menstrual irregularity [24], infertility [25], and dysmenorrhea [26], and has been reported to prevent and treat pain and improve the QOL by supplementing a weakened Qi and blood in patients with postpartum body pain [27,28].

In this systematic review, we analyzed the studies on the effectiveness and safety of herbal medicines for postpartum pain to improve maternal health.

## 2. Materials and Methods

### 2.1. Protocol and Registration

This systematic review protocol was registered in the International Prospective Register of Systematic Reviews under the registration number PROSPERO 2022:CRD42022326696. It can be accessed at https://www.crd.york.ac.uk/prospero/display_record.php?ID=CRD42022326696 accessed on 10 May 2022.

### 2.2. Data Sources and Searches

We searched through PubMed, EMBASE, Cochrane Library, Chinese databases (China National Knowledge Infrastructure (CNKI)), Japanese databases (Citation Information by NII (CiNii)), and Korean medical databases (Korea Studies Information Service System, Oriental Medicine Advanced Searching Integrated System, ScienceOn, and Korean Medical Database).

The search terms were as follows: “postpartum” OR “postnatal” OR “puerperal” OR “after childbirth” AND “pain” OR “pantalgia” OR “ache” OR “soreness” OR “arthrodynia” OR “bodily pain” OR “myalgia” OR “arthralgia” OR “paresthesia” OR “hypesthesia” OR “dysesthesia” OR “hypoesthesia” OR “numbness” OR “tingling” OR “cold hypersensitivity” OR “cold sense” OR “cold intolerance” OR “sensation disorder” OR “sensation dysfunction” OR “sensation loss” AND “Medicine, Chinese Traditional” OR “Medicine, Korean Traditional” OR “Medicine, Kampo” OR “Herbal Medicine” OR “Plants, Medicinal” OR “Drugs, Chinese Herbal” OR “herbal drug” OR “Chinese formula” OR “Chinese prescription” OR “Chinese decoction” OR “phytomedicine” OR “herb” OR “botanic”. These terms were modified based on an electronic search database. The search strategy for each database is presented in Appendix A.

### 2.3. Study Selection

#### 2.3.1. Types of Studies

All randomized controlled trials (RCTs) that assessed the effects of herbal medicines on puerperal wind syndrome were included. We excluded non-randomized trials, including clinical studies (case studies, case series, and case-controlled trials), experimental studies, animal studies, surveys, and reviews.

#### 2.3.2. Participants

We included all patients diagnosed with puerperal wind syndrome suffering from pain, paresthesia, and dysesthesia, regardless of age.

#### 2.3.3. Types of Interventions

Any herbal medicine administered orally was included.

#### 2.3.4. Types of Comparisons

We compared herbal medicine with no treatment, a placebo or sham treatment, or conventional treatment. Furthermore, we included RCTs comparing herbal medicine plus conventional treatment with conventional treatment and herbal medicine plus traditional Korean treatment (e.g., moxibustion, warm needle acupuncture, or Chuna) with traditional Korean treatment.

#### 2.3.5. Types of Outcome Measures

In this study, the primary outcomes were the pain score and rate of effectiveness for treating puerperal wind syndrome. The efficacy rate was defined as the number of patients whose symptoms improved among all patients. Secondary outcomes included QOL, functionality scores, and adverse events.

### 2.4. Data Extraction

Using a predefined data extraction form, three authors (H.-Y. Lee, S.-I. Hwang, and N.-Y. Kwon) independently extracted the data. The three independent reviewers (H.-Y. Lee, S.-I. Hwang, and N.-Y. Kwon) collected data with regard to the author’s information, sample size, interventions, outcome measures, main results, and adverse events. Regarding herbal medicine interventions, we collected the following data: name of herbal medicine, composition of herbal medicine, modified herbs, and duration and frequency of herbal medicine use. Any disagreements regarding article selection were resolved by a discussion with the third author (J.-K. Park).

### 2.5. Assessment of Risk of Bias (ROB)

Using the Cochrane Collaboration’s ROB tool [29], two independent researchers (S.-H. Sung and S.-J. Cho) evaluated the ROB for the included RCTs. The Cochrane Collaboration tool comprises seven domains; however, we evaluated the following six assessment methods: (1) random sequence generation, (2) allocation concealment, (3) blinding of participants, (4) blinding of assessors, (5) incomplete outcome data, and (6) selective outcome reporting. For each domain, the ROB was rated as low risk (L), high risk (H), or unclear (U). Different opinions on scoring were resolved through a discussion with the third author (Y.-J. Yoon).

### 2.6. Data Analyses

Statistical analyses were carried out utilizing RevMan 5.4 (version 5.4 for Windows (Nordic Cochrane Center, Copenhagen, Denmark)). With 95% confidence intervals, the continuous and dichotomous data are reported as mean differences and risk ratios, respectively. The inter-study heterogeneity was evaluated using the I^2^ test, with I^2^ values of 0–40%, 30–60%, 50–90%, and 75–100% representing absence or mild, moderate, substantial, and full heterogeneity, respectively [30]. When the I^2^ values were 50% and >50%, respectively, fixed and random effects models were applied, and a subgroup analysis was conducted to identify the possible reasons for heterogeneity [30]. A sensitivity analysis was planned using trials with a low ROB to investigate the possible contribution of methodological quality. If a meta-analysis could not possible because of the considerable variation in the study characteristics, a summary of the findings is discussed in Section 3.

## 3. Results

### 3.1. Study Selection and Description

The database searches identified 1497 potentially related studies, and we identified 3 articles through other sources. After excluding 99 duplicated articles, 246 RCTs were considered for full-article assessment by reviewing titles and abstracts. Of them, 171 were not RCTs, 55 included interventions that were not herbal medicine, and 11 were unqualified control interventions; all of them were excluded for eligibility. In the end, nine RCTs were included in our review. All the studies [31,32,33,34,35,36,37,38,39] were conducted in China and published in Chinese. Figure 1 presents a flowchart of the study selection process recommended by the Preferred Reporting Items for Systematic Reviews and Meta-Analyses guidelines [40]. The specifics of the included studies are summarized in Table 1.

### 3.2. Participants

In total, 652 patients with puerperal wind syndrome were included. The experimental and control groups included 326 patients each. The final analyses included 324 and 323 patients from the experimental and control groups, respectively.

### 3.3. Intervention

We compared herbal medicine with conventional treatment [31,32,33,34,35], herbal medicine in combination with conventional treatment versus conventional treatment alone [36], and herbal medicine in combination with other traditional Korean treatments (e.g., moxibustion, warm needle acupuncture, or Chuna) versus other traditional Korean treatments alone [37,38,39]. Five RCTs compared herbal medicines with conventional treatment [31,32,33,34,35]. One study compared herbal medicine plus indomethacin cataplasm and indomethacin cataplasm alone [36]. One study compared herbal medicine plus warm needle acupuncture and warm needle acupuncture alone [37], herbal medicine plus moxibustion and moxibustion alone [38], and herbal medicine plus Chuna and Chuna alone [39]. Table 2 and Table 3 shows the characteristics of the herbal and conventional medicine interventions included in the RCTs.

#### 3.3.1. Name of Prescription

The prescription names were noted in all the studies. The Wenjing decoction [31,32] and Chanhoubi decoction [35,36] were included in two studies. In addition, the Xiaoxuming Tang [33], Huangqi Guizhi Wuwu [34], Yangyuan Huoxue [37], Duhuo Jisheng [38], and Duhuo Jisheng decoctions [39] were used in one study.

#### 3.3.2. Herbs Used in the Included Studies

A total of nine studies used 50 herbs. The most frequently used herbs were *Angelicae Gigantis Radix* [31,32,34,35,36,37,38,39], *Glycyrrhizae Radix et Rhizoma* [31,32,33,35,36,37,38,39], and *Paeoniae Radix Alba* [31,32,34,35,36,37,38,39], which were used in eight studies; *Asiasari Radix et Rhizoma* was used in seven studies [31,32,35,36,37,38,39]; *Cinnamomi Ramulus* [31,32,34,35,36,37] and *Cnidii Rhizoma* [31,32,33,37,38,39] were used in six studies; *Acontii Lateralis Radix Preparata* [31,32,33,35,37] and *Astragali Radix* [31,32,34,35,36] were used in five studies; *Achyranthis Radix* [34,37,38,39], *Moutan Cortex* [31,32,35,36], *Poria Sclerotium* [35,36,38,39], and *Saposhnikoviae Radix* [33,34,38,39] were used in four studies; *Araliae Continentalis Radix* [37,38,39], *Asini Corii Colla* [31,32,37], *Atractylodis Rhizoma Alba* [31,32,34], *Cinnamomi Cortex* [33,38,39], *Eucommiae Cortex* [37,38,39], *Gingseng Radix* [31,32,38], *Liriopis Tuber* [31,32,37], and *Zingiberis Rhizoma Recens* [31,32,34] were used in three studies (Figure 2).

#### 3.3.3. Modified Herbs

In four studies, the herbs were modified according to symptoms [31,32,34,37]. The first medicinal herb was added according to the part of the body where the pain was severe. For lower extremity pain, *Achyranthis Radix* [31,32], *Araliae Continentalis Radix*, *Sinomeni Caulis et Rhizoma*, and *Piperis Kandsurae Caulis* were applied [34]. In the case of upper extremity pain, *Osterici Radix*, *Curcumae Longae Rhizoma*, and *Gentianae Macrophyllae Radix* were added [34]. In cases of lower back pain [31,32], *Loranthi Ramulus*, *Dipsaci Radix*, and *Eucommiae Cortex* were added. In cases of chronic impediment disease [34], *Scolopendrae Corpus* and *Asiasari Radix et Rhizoma* were added. In cases of headaches, *Cnidii Rhizoma* was added [31,32]. Considering the accompanying systemic symptoms, if a patient complained of aversion to wind, *Araliae Continentalis Radix*, *Aconiti Tuber*, and *Saposhnikoviae Radix* [31] were added, and *Cervi Cornus Colla* [32] was added for aversion to cold. In the case of spontaneous sweating, *Trtici Levis Semen* and *Ephedrae Radix et Rhizoma* [34] were added; in the case of sleep inability, *Zizyphi Spinosi Semen* and *Polygoni Multiflori Caulis* [37] were added; and in the case of reduced food intake, *Galli Stomachichum Corium* [37] was added (Table 2).

#### 3.3.4. Duration and Frequency of Taking Herbal Medicine

The duration of herbal medicine use varied from 7 days to 3 months. In three studies, most participants took herbal medicine for 1 month [31,32,34]. The frequency of herbal medicine use was twice a day in five studies [31,32,33,38,39], three times a day in two studies, and once a day in two studies [34,35] (Table 2).

#### 3.3.5. Formulation of Herbal Medicine

The formulation of herbal medicine was a decoction in all nine studies.

### 3.4. Outcomes

#### 3.4.1. Pain (Visual Analog Scale [VAS])

(1)Herbal medicine versus conventional treatment

We analyzed three RCTs [31,34,35] comparing the degree of pain improvement using VAS between the herbal and Western medicine treatments. The pain reduction effect of the herbal medicine treatment group was statistically better than that of the Western medicine treatment group (n = 217, risk ratio (RR): −1.40, 95% CI −2.02 to −0.78, *p* < 0.00001; Figure 3).

(2)Herbal medicine plus warm needle acupuncture versus warm needle acupuncture

One study [37] reported a VAS score comparing herbal medicine plus warm needle acupuncture with warm needle acupuncture alone; herbal medicine plus warm needle acupuncture significantly improved the VAS score (*p* < 0.00001).

(3)Herbal medicine plus moxibustion versus moxibustion

One study [38] compared herbal medicine plus moxibustion with moxibustion alone and found that herbal medicine plus moxibustion resulted in a significantly better improvement in the VAS score (*p* < 0.05).

(4)Herbal medicine plus Chuna versus Chuna

One RCT [39] compared herbal medicine plus Chuna with Chuna and found that herbal medicine plus Chuna significantly improved the VAS score (*p* < 0.0001).

#### 3.4.2. Total Effective Rate

(1)Herbal medicine versus conventional treatment

Five RCTs compared the treatment effects of herbal medicine and Western medicine using the total effective rate [31,32,33,34,35]; the total effective rate of the herbal medicine group was statistically higher than that of the Western medicine treatment group (n = 369, RR 1.19, 95% CI 1.10 to 1.29, *p* < 0.0001, Figure 4, Table 4).

(2)Herbal medicine plus warm needle acupuncture versus warm needle acupuncture

One study [37] reported that the total effective rate for puerperal wind syndrome was higher in the treatment group than in the control group; however, the difference was not significant (*p* = 0.11), (Table 4).

(3)Herbal medicine plus moxibustion versus moxibustion

One study [38] compared herbal medicine plus moxibustion with moxibustion alone and found that herbal medicine plus moxibustion was more effective than moxibustion in the control group; however, there was no significant difference in the total effective rate (*p* = 0.28), (Table 4).

(4)Herbal medicine plus Chuna versus Chuna

One RCT [39] compared herbal medicine plus Chuna with Chuna alone and found that herbal medicine plus Chuna significantly improved the total effective rate (*p* < 0.05), (Table 4).

#### 3.4.3. Scores of TCM syndromes

Three RCTs [31,34,35] comparing herbal medicine with conventional treatment reported significant effects in the herbal medicine group with respect to TCM syndrome scores for puerperal wind syndrome (n = 217, RR −4.05, 95% CI −6.05 to −2.05, *p* < 0.0001, Figure 5).

#### 3.4.4. Oswestry Disability Index (ODI)

One RCT [39] comparing herbal medicine plus Chuna versus Chuna alone reported that herbal medicine plus Chuna significantly improved the ODI scores (*p* < 0.0001).

#### 3.4.5. QOL

(1)Herbal medicine versus conventional treatment

Three RCTs [31,33,35] were used to evaluate the QOL. The 36-Item Short-Form Survey (SF-36) was used in two studies [31,33], and the World Health Organization Quality of Life Scale (WHOQOL-BREF) was used in one study [35]. Lu’s RCT [25] used SF-36 and physical functioning, bodily pain, social functioning, and role-emotional as indicators, and the treatment group showed significantly improved results (*p* < 0.00001). Chen’s RCT [33] also used SF-36, role-physical (*p* < 0.001), general health (*p* = 0.002), vitality (*p* ≤ 0.0001), social functioning (*p* < 0.002), role-emotional (*p* = 0.0008), and mental health (*p* = 0.001), and all of them were more effective in the treatment group than in the control group. One RCT [35] reported significant effects with respect to the WHOQOL-BRE for puerperal wind syndrome (*p* < 0.05).

As a result of a meta-analysis of two RCTs [31,33] comparing herbal medicine versus conventional treatment, the role-emotional indicator of the herbal medicine group was significantly improved (n = 172, RR 14.33, 95% CI 13.13 to 15.53, *p* < 0.00001, Figure 6), and social functioning was also improved; however, there was no statistical significance (n = 172, RR 11.14, 95% CI −2.21 to 24.50, *p* = 0.10, Figure 7).

(2)Herbal medicine plus conventional treatment versus conventional treatment alone

One study [36] reported that herbal medicine plus conventional treatment significantly improved QOL scores (*p* < 0.00001).

(3)Herbal medicine plus warm needle acupuncture versus warm needle acupuncture

In one study [37] comparing herbal medicine plus warm needle acupuncture versus warm needle acupuncture alone, herbal medicine plus warm needle acupuncture significantly improved the SF-36 scores (*p* < 0.0001).

#### 3.4.6. Adverse Events

Mild adverse events were reported in five studies [31,32,33,35,39]. The adverse events in the treatment group included nausea, vomiting, gastrointestinal discomfort, headaches, pain, and fatigue. The pooled effects of the five studies were higher in the control group than in the experimental group; however, they were not statistically significant (n = 350, risk difference −0.06, 95% CI −0.13 to 0.01, *p* = 0.07, I^2^ = 35%, Figure 8).

## 4. Assessment of ROB

The ROB of the included RCTs is presented in Figure 9. Regarding the randomization procedure, six studies [32,33,34,37,38,39] mentioned adequate methods of randomization concealment. Furthermore, two RCTs [35,36] did not report random sequence generation methods, and the remaining study [31] was considered to have a high ROB as it conducted a controlled study according to visit numbers. Odd numbers were assigned to the control group and even numbers to the treatment group. None of the included studies mentioned the proper allocation concealment method. One study [31] reported an inadequate method of allocation concealment using visit numbers.

None of the included RCTs described the blinding of participants and outcome assessors. All the included clinical studies had a low ROB in addressing incomplete outcome data: eight RCTs [31,32,33,34,36,37,38,39] had no missing outcome data, and another study [35] had missing outcome data; however, the dropout rate did not exceed 20% for short-term and 30% for long-term follow-ups. For selective reporting, none of the trials registered study protocols before conducting the clinical studies. Two studies [36,37] evaluated all research methods and included a scale that could evaluate postpartum pain. 

## 5. Publication Bias

A funnel plot of the primary outcome (the total effectiveness rate for puerperal wind syndrome) was constructed. No significant asymmetry was observed on visual inspection of the funnel plot (Figure 10).

## 6. Discussion

In this study, we analyzed nine RCTs on the efficacy and safety of herbal medicines in patients with postpartum body pain. An analysis of the included studies showed that herbal medicine improved the VAS score, total effective rate, scores of TCM syndrome, ODI, and QOL of patients with postpartum body pain (Figure 11). The Wenjing decoction significantly improved the VAS score, scores of TCM syndrome, and SF-36 scores compared with ibuprofen, while the Chanhoubi, Xiaoxuming Tang, and Huangqi Guizhi Wuwu decoctions had statistically significant effects on the pain index and QOL compared to indomethacin. According to TCM, post-delivery is the period where one should be cautious of food intake, which burdens the digestive function because the spleen and stomach functions are weakened due to deficiencies in vital energy (Qi) and blood [41]. NSAIDs, which are widely used for pain relief, often cause side effects such as nausea, vomiting, and gastrointestinal discomfort [15]. In this review, the side effects in the herbal medicine group were significantly lower than those in the control group, and all were minor. Patients who choose herbal medicines for pain treatment desire to benefit from a lower risk of side effects [42]. Therefore, administering herbal medicines to patients with postpartum body pain may be more effective and safer than administering NSAIDs, and herbal medicine may be the preferred treatment for patients who cannot take NSAIDs owing to their side effects.

On comparing RCTs in which the Yangyuan Huoxue, Duhuo Jisheng, and Duhuo Jisheng decoctions were applied together with other TCM treatments such as warm needle acupuncture, moxibustion, Chuna, and other treatments alone, the pain-related index and QOL were significantly improved in the combined treatment group. In the treatment of pain in TCM, combining herbal medicine with acupuncture was more effective than acupuncture alone [43]. Postpartum pain intensity is significantly related to maternal parenting self-efficacy at 3 months postpartum [13]; thus, postpartum pain requires early treatment not only for improving maternal QOL but also for newborn care. Furthermore, the participation of women in society has increased since the 20th century, and women return to work 12 weeks after giving birth in general, with 23% of American women returning to work even earlier, within 10 days of delivery [44,45]. The physical and mental health of mothers who simultaneously manage a job and childrearing are relatively poorer than those who take maternity leave, which reduces their QOL and causes them to abandon work [46]. A combination treatment in which herbal medicine is administered with other TCM treatments can effectively help in the recovery from physical symptoms; therefore, it is expected to improve maternal health, satisfaction with social roles, and overall QOL.

In all the included studies, the most commonly used medicinal plants were *Angelicae gigantis Radix*, *Glycyrrhizae Radix et Rhizoma*, *Paeoniae Radix Alba*, and *Asiasari Radix et Rhizoma* (Figure 11). *Angelicae gigantis Radix*, whose active ingredient is decursin, replenishes the blood, promotes blood flow, and reduces pain [47,48,49]. *Glycyrrhizae Radix et Rhizoma*, whose active ingredient is glycyrrhizin, also alleviates pain and has immunoregulatory effects [50,51]. *Paeoniae Radix Alba* invigorates blood circulation so has been used to treat gynecological problems, pain, cramp, and congestion [52,53]. Paeoniflorin, its active ingredient, has anti-inflammatory immune regulatory effects and antioxidant properties [54]. *Asiasari Radix et Rhizoma*, whose active ingredient is methyleugenol, has anti-inflammatory effects and antinociceptive effects [55].

All four of the above active ingredients have anti-inflammatory effects, which are characterized by controlling the signaling pathway and inhibiting inflammatory mediators such as prostaglandin and interleukin, and are widely used in TCM as drugs to suppress pain.

Safety assessment is very important because drug use during breastfeeding can affect mothers or infants. Previous studies that evaluated the safety of taking herbs included self-report surveys [56,57], cross-sectional studies [58], and review papers [59], and neither maternal nor natal adverse effects were reported. Several herbs were used as galactagogues and were effective, so mothers with insufficient breast milk were taking herbs to see the effect. The adverse events identified in this paper were mild side effects such as pain, nausea, and fatigue, and there was no significant difference from the control group; therefore, it was difficult to see them as being caused by herbal medicine.

During the postpartum period, symptoms such as pain in vulnerable mothers are evaluated, and customized management and care are required. In TCM, in the puerperium, the vital energy (Qi) and blood are lost due to bleeding and overwork during childbirth, making the mother weak and allowing pathogens to easily enter the body from the outside; thus, body pain is likely to occur. Therefore, in the treatment of postpartum body pain, tonifying the Qi and replenishing blood are the main treatment strategies [60,61]. Although herbal medicine for pain is not the most powerful analgesic, it is known to have beneficial effects on mild-to-moderate pain [42]. Additionally, herbal medicines combined with various medicinal plants exhibit anti-fatigue activity by affecting diverse targets through multiple pathways [62]. Herbal medicine has been reported to improve QOL by not only improving body pain but also significantly improving the overall health and vitality of postpartum mothers [28,63]. Therefore, as herbal medicine enhances vitality, improves pain, and helps improve maternal health, it is considered a relatively safe treatment for exhausted mothers who complain of body pain.

This study’s limitations include its focus on the East Asian concept of puerperal wind, leading to the inclusion of studies solely from China, with limited RCTs available. Randomization, allocation, and concealment were not mentioned in any of the studies; therefore, the possibility of bias was high. Due to the limited number of studies, the grades of evidence were evaluated as moderate or low (Table 5, Table 6, Table 7, Table 8 and Table 9). Herbal medicine is administered based on the symptoms of the patient and the pattern of differentiation in the diagnosis. Because of the unique characteristics of TCM, the prescriptions administered were discordant between each study, and the modified medicinal plants according to the patient’s symptoms were different. Therefore, it was difficult to provide a standardized herbal medicine prescription for puerperal wind syndrome. We considered subgroup analysis for different forms, timing, doses, and duration of use of different herbal medicines. However, the number of papers included in this study (nine) was too small to conduct a subgroup analysis. Also, the compositions of the herbal decoctions used were all different, so an additional subgroup analysis could not be performed.

Nevertheless, we searched several literature databases using a comprehensive search strategy and screened the studies published in English, Korean, and Chinese without limiting the search to any particular language. Moreover, this is the first systematic review of the effects of herbal medicines on puerperal wind syndrome. Future studies with a larger sample size involving allocation, concealment, and blinding of participants, examiners, and assessors are required. We expect that more RCTs of herbal medicine for postpartum pain will be conducted in the future and that a subgroup analysis will be performed in an updated systematic review. Furthermore, an RCT is needed to compare the effects of herbal medicine prescriptions by administering herbal medicines based on the pattern of differentiation in the diagnosis in patients with puerperal wind syndrome. As mothers often experience fatigue, poor sleep, and digestive disorders after childbirth, a study design that can present standardized evidence for the modification of additional symptoms reported by patients should be developed. In addition, research is necessary to observe the long-term QOL and health of the children of women treated with herbal medicine for postpartum pain.

## 7. Conclusions

Based on the evidence from this systematic review, herbal medicine treatment showed better improvement in clinical efficacy, pain index, ODI, and QOL and fewer side effects than conventional treatment and other TCM treatments alone for puerperal wind syndrome. Therefore, herbal medicine is considered a complementary and integrated treatment option for postpartum pain to improve postpartum maternal health and long-term QOL in women. In the future, it will be necessary to minimize the ROB in a large sample of patients and evaluate the treatment effects according to the pattern of differentiation in the diagnosis criteria. In addition, clinical follow-up studies are needed to observe the long-term effects of herbal medicine administration on postpartum pain, QOL, and childrearing.

## Figures and Tables

**Figure 1 healthcare-11-02743-f001:**
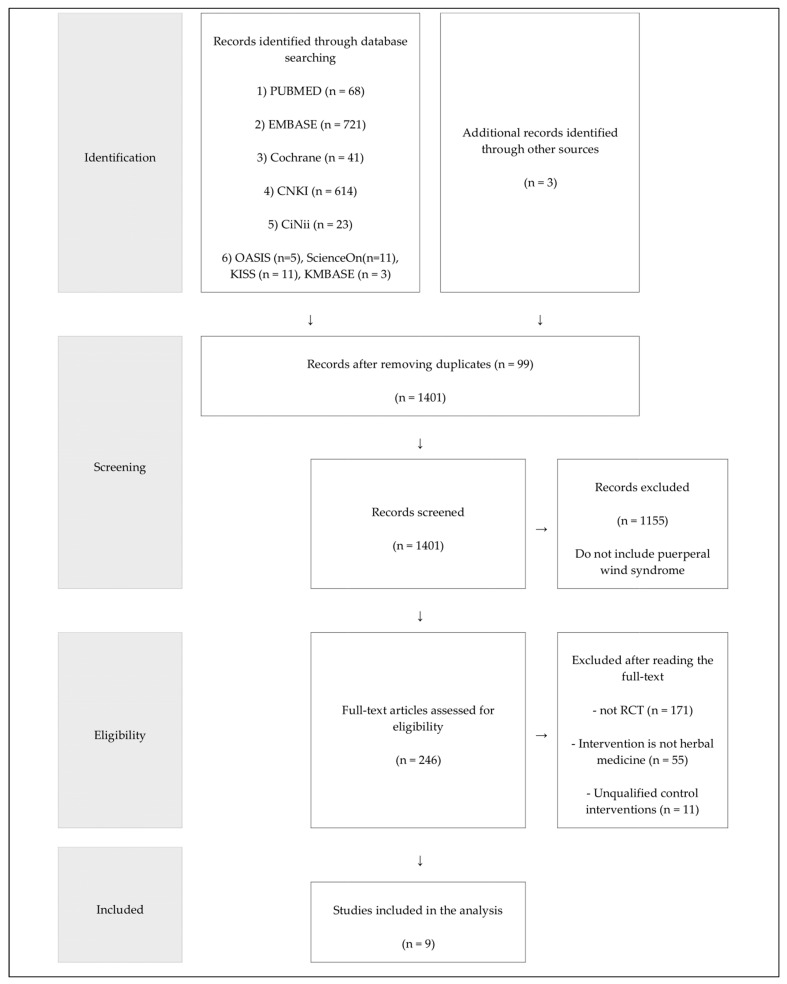
Flowchart of the RCT selection process. CCTs, controlled clinical trials; RCTs, randomized controlled trials; CNKI, China National Knowledge Infrastructure; CiNii, Citation Information by NII; KISS, Korea Studies Information Service System; OASIS, Oriental Medicine Advanced Searching Integrated System; KMBASE, Korean Medical Database.

**Figure 2 healthcare-11-02743-f002:**
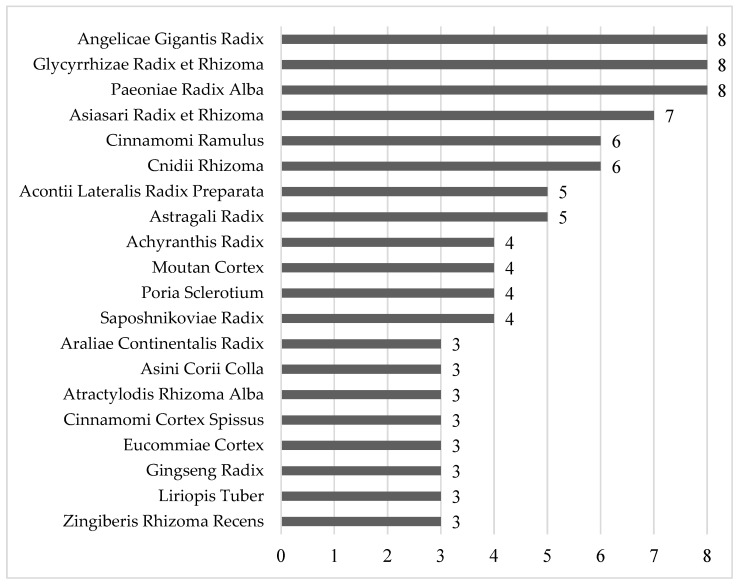
Frequency of herbs used in the included studies.

**Figure 3 healthcare-11-02743-f003:**
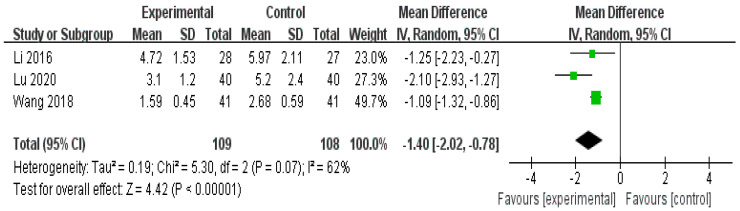
Forest plot for visual analog score (VAS). VAS of the herbal medicine treatment group was significantly lower than that of the Western medicine treatment group [31,34,35].

**Figure 4 healthcare-11-02743-f004:**
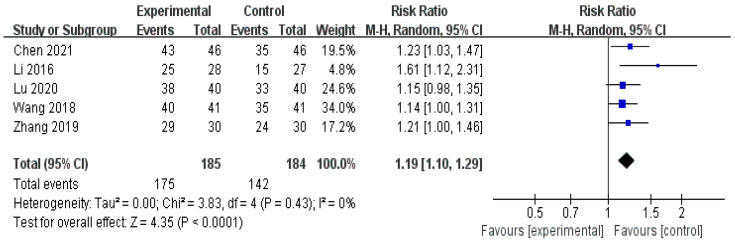
Forest plot for total effective rate. CI, confidence interval; M-H, Mantel–Haenszel Formula. The total effective rate of the herbal medicine group was significantly higher than that of the Western medicine treatment group [31,32,33,34,35].

**Figure 5 healthcare-11-02743-f005:**
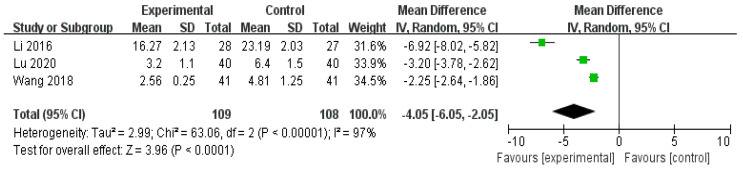
Forest plot for scores of Traditional Chinese Medicine (TCM) syndromes. CI, confidence interval. The TCM of the herbal medicine group was statistically lower than that of the Western medicine treatment group [31,34,35].

**Figure 6 healthcare-11-02743-f006:**
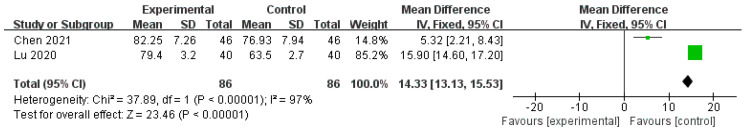
Forest plot for SF-36 (role-emotional). CI, confidence interval. The role-emotional indicator of the herbal medicine group was significantly improved compared to conventional treatment [31,33].

**Figure 7 healthcare-11-02743-f007:**
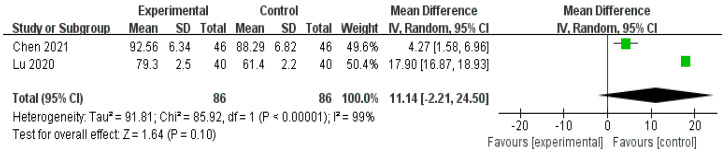
Forest plot for SF-36 (social functioning). CI, confidence interval. The social functioning of the herbal medicine group had no statistical significance compared to conventional treatment [31,33].

**Figure 8 healthcare-11-02743-f008:**
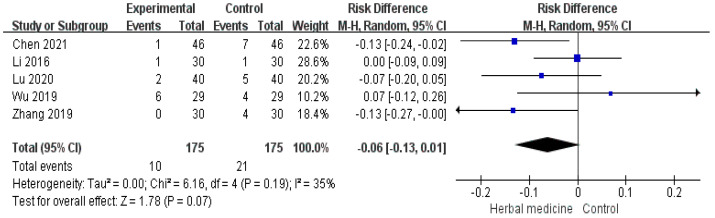
Forest plot for adverse events. CI, confidence interval; M-H, Mantel–Haenszel Formula. Number of adverse events was not statistically significant between both groups [31,32,33,35,39].

**Figure 9 healthcare-11-02743-f009:**
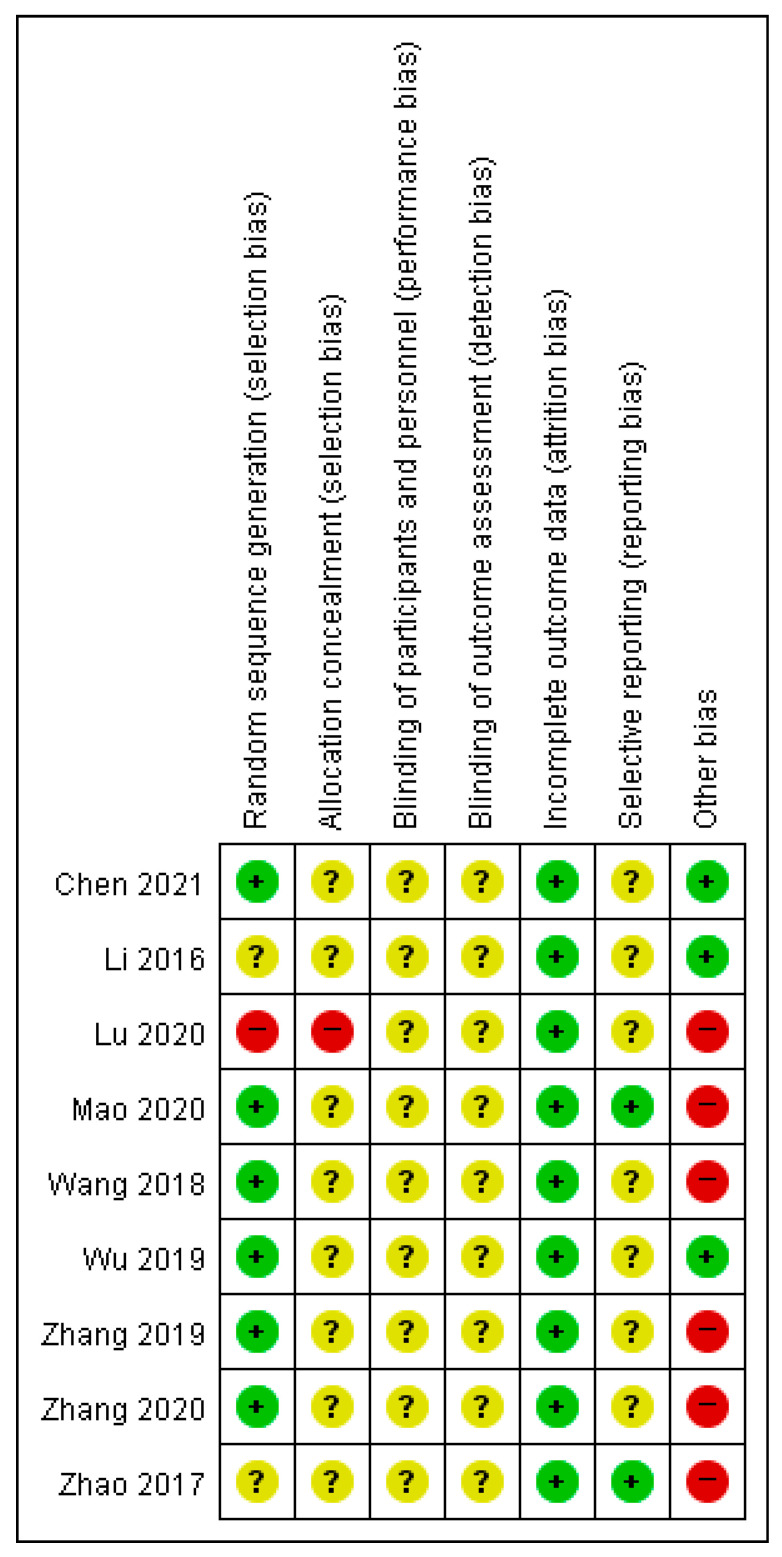
Risk of bias graph and summary [31,32,33,34,35,36,37,38,39]. +, Low risk of bias; ?, Unclear risk of bias; −, High risk of bias.

**Figure 10 healthcare-11-02743-f010:**
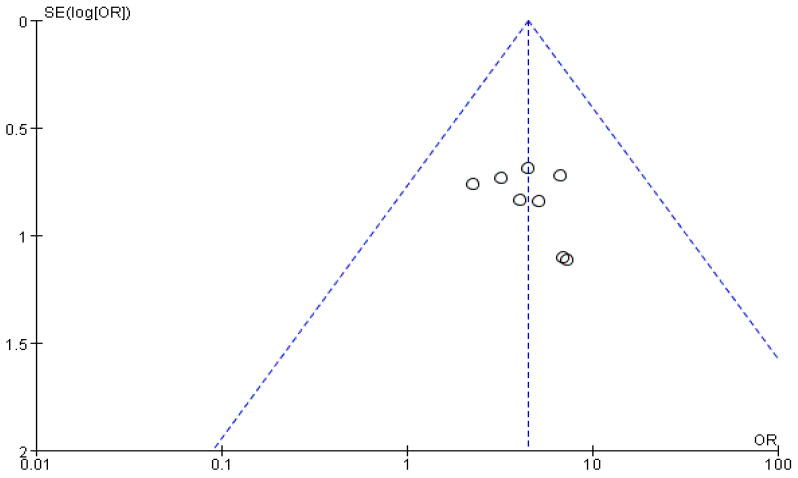
Funnel plot: total effectiveness rate (TER) for puerperal wind syndrome.

**Figure 11 healthcare-11-02743-f011:**
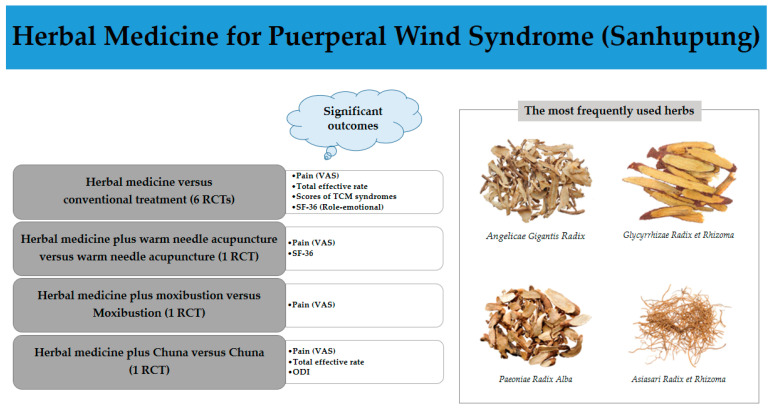
Summary of overall findings from this systematic review.

**Table 1 healthcare-11-02743-t001:** Characteristics of the included studies.

Author, Year	Sample Size	Average Age (Experimental Group/Control Group)	Primipara/Multipara (Experimental Group/Control Group)	Experimental Group (No. of Participants Analyzed/ Randomized)	Control Group (No. of Participants Analyzed/ Randomized)	Outcome Measures	Main Results	Adverse Events
Lu (2020) [31]	80	27.5 ± 2.4 28.2 ± 3.2	23/17 21/19	Modified Wenjing decoction (40/40)	Ibuprofen (40/40)	(1) TER	(1) No significant difference (*p* = 0.08)	TG < CG (*p* < 0.05)
						(2) VAS	(2) Positive (*p* < 0.05)	
						(3) Scores of TCM syndromes	(3) Positive (*p* < 0.05)	
						(4) SF-36 ① Physical functioning ② Bodily pain ③ Social functioning ④ Role-emotional	(4) ① Positive (*p* < 0.00001) ② Positive (*p* < 0.00001) ③ Positive (*p* < 0.00001) ④ Positive (p < 0.00001)	
Zhang (2019) [32]	60	29.3 29.2	18/12 19/11	Wenjing decoction (30/30)	Ibuprofen (30/30)	(1) TER	(1) No significant difference (*p* = 0.05)	TG < CG (*p* < 0.05)
Chen (2021) [33]	92	33.29 ± 3.45 33.09 ± 3.31	12/34 13/33	Xiaoxuming Tang (46/46)	Indomethacin cataplasm (46/46)	(1) TER	(1) Positive (*p* < 0.05)	TG < CG (*p* < 0.05)
						(2) SF-36 ① Role-physical ② General health ③ Vitality ④ Social Functioning ⑤ Role-emotional ⑥ Mental health	(2) ① Positive (*p* = 0.001) ② Positive (*p* = 0.002) ③ Positive (*p* < 0.0001) ④ Positive (*p* = 0.002) ⑤ Positive (*p* = 0.0008) ⑥ Positive (*p* = 0.001)	
Wang (2018) [34]	82	29.02 ± 4.41 28.79 ± 4.25	31/10 33/8	Huangqi Guizhi Wuwu decoction (41/41)	Indomethacin cataplasm (41/41)	(1) TER	(1) No significant difference (*p* = 0.05)	N.R.
						(2) VAS	(2) Positive (*p* < 0.05)	
						(3) Scores of TCM syndromes	(3) Positive (*p* < 0.05)	
						(4) PSQI	(4) Positive (*p* < 0.05)	
						(5) Number of painful joints	(5) Positive (*p* < 0.05)	
						(6) Duration of morning stiffness	(6) Positive (*p* < 0.05)	
Li (2016) [35]	60	30.16 ± 7.87 29.73 ± 7.49	N.R.	Chanhoubi decoction (28/30)	Indomethacin cataplasm (27/30)	(1) TER	(1) Positive (*p* < 0.05)	No significant difference
						(2) VAS	(2) Positive (*p* < 0.05)	
						(3) Scores of TCM syndromes	(3) Positive (*p* < 0.01)	
						(4) WHOQOL-BREF	(4) Positive (*p* < 0.05)	
						(5) Number of painful joints	(5) No significant difference	
						(6) Duration of morning stiffness	(6) Positive (*p* < 0.01)	
Zhao (2018) [36]	96	N.R.	N.R.	Chanhoubi decoction + indomethacin cataplasm (48/48)	Indomethacin cataplasm (48/48)	(1) Pain improvement rate	(1) No significant difference (*p* = 0.31)	N.R.
						(2) Period to improvement in symptoms (weeks)	(2) Positive (*p* < 0.00001)	
						(3) Scores of quality of life	(3) Positive (*p* < 0.00001)	
Mao (2020) [37]	64	28.34 ± 3.21 28.14 ± 3.17	N.R.	Yangyuan Huoxue decoction + warm needle acupuncture (32/32)	Warm needle acupuncture (32/32)	(1) TER	(1) No significant difference (*p* = 0.11)	N.R.
						(2) VAS	(2) Positive (*p* < 0.00001)	
						(3) ODI	(3) Positive (*p* < 0.00001)	
						(4) SF-36 ① Sleep ② Mental health ③ Diet ④ Social functioning	(4) ① Positive (*p* < 0.00001) ② Positive (*p* < 0.00001) ③ Positive (*p* < 0.00001) ④ Positive (*p* < 0.00001)	
						(5) Serum cytokine level ① Serum 6-Keto-PGFlα ② IL-1β ③ CGRP ④ TXB2	(5) ① Positive (*p* = 0.004) ② Positive (*p* < 0.00001) ③ Positive (*p* < 0.00001) ④ Positive (*p* < 0.00001)	
Zhang (2020) [38]	60	27.1 ± 4.0 26.3 ± 3.7	N.R.	Duhuo Jisheng decoction + moxibustion (30/30)	Moxibustion (30/30)	(1) TER	(1) No significant difference (*p* = 0.28)	N.R.
						(2) VAS	(2) Positive (*p* = 0.0006)	
Wu (2019) [39]	58	35.41 ± 5.24 36.02 ± 5.09	N.R.	Duhuo Jisheng decoction + Chuna (29/29)	Chuna (29/29)	(1) TER	(1) Positive (*p* = 0.04)	TG < CG (*p* < 0.05)
						(2) VAS	(2) Positive (*p* < 0.0001)	
						(3) ODI	(3) Positive (*p* < 0.0001)	
						(4) Recurrence rate	(4) Positive (*p* = 0.04)	

TER, total effectiveness rate; TG, treatment group; CG, control group; VAS, visual analog scale; N.R., not recorded; PSQI, Pittsburgh Sleep Quality Index; WHOQOL-BREF, World Health Organization Quality of Life Scale; SF-36, 36-Item Short-Form Survey; ODI, Oswestry Disability Index; Serum 6-Keto-PGF1α, serum ketoprostaglandin Flα; IL-1β, interleukin-1β; CGRP, calcitonin gene-related peptide; TXB2, Thromboxane 2; TCM, Traditional Chinese Medicine.

**Table 2 healthcare-11-02743-t002:** Characteristics of herbal medicine interventions in the included studies.

Author, Year	Duration	Frequency	Composition of Herbal Medicine	Modified Herb
Lu (2020) [31]	1 month	Twice a day	*Glycyrrhizae Radix et Rhizoma* 6 g, *Zingiberis Rhizoma Recens* 3 pieces, *Asini Corii Colla* 10 g, *Moutan Cortex* 8 g, *Liriopis Tuber* 20 g, *Asiasari Radix et Rhizoma* 1 g, *Cnidii Rhizoma* 10 g, *Angelicae Gigantis Radix* 15 g, *Schizonepetae Spica* 15 g, *Evodiae Fructus* 8 g, *Cinnamomi Ramulus* 15 g, *Atractylodis Rhizoma alba* 10 g, *Paeoniae Radix Alba* 30 g, *Acontii Lateralis Radix Preparata* 8 g, *Astragali Radix* 15 g, *Astragali Radix Preparata* 20 g, *Gingseng Radix* 8 g	- Lower extremity pain: Achyranthis Radix 15 g - Aversion to cold: Cervi Cornus Colla 15 g, Ginseng Radix change to Red Ginseng - Lower back pain: Loranthi Ramulus 15 g, Dipsaci Radix 15 g, Eucommiae Cortex 15 g - Headache: Cnidii Rhizoma increased to 15 g - Aversion to wind: Araliae Continentalis Radix 6 g, Aconiti Tuber 6 g, Saposhnikoviae Radix 10 g
Zhang (2019) [32]	1 month	Twice a day	*Astragali Radix* 15 g, *Angelicae Gigantis Radix* 15 g, *Schizonepetae Spica* 15 g, *Astragali Radix Preparata* 20 g, *Liriopis Tuber* 20 g, *Gingseng Radix* 8 g, *Acontii Lateralis Radix Preparata* 8 g, *Moutan Cortex* 8 g, *Evodiae Fructus* 8 g, *Paeoniae Radix Alba* 30 g, *Asini Corii Colla* 10 g, *Atractylodis Rhizoma alba* 10 g, *Cnidii Rhizoma* 10 g, *Cinnamomi Ramulus* 15 g, *Asiasari Radix et Rhizoma* 1 g, *Glycyrrhizae Radix et Rhizoma* 6 g, *Zingiberis Rhizoma Recens* 3 pieces	- Lower back pain: Dipsaci Radix 15 g, Loranthi Ramulus 15 g, Eucommiae Cortex 15 g - Headache: Cnidii Rhizoma 5 g - Lower extremity pain: Achyranthis Radix 15 g - Aversion to cold: Cervi Cornus Colla 15 g
Chen (2021) [33]	3 months	Twice a day	*Sinomeni Caulis et Rhizoma* 10 g, *Paeoniae Radix rubra* 10 g, *Armeniacae Semen* 10 g, *Saposhnikoviae Radix* 10 g, *Cnidii Rhizoma* 10 g, *Cinnamomi Cortex* 15 g, *Scutellariae Radix* 15 g, *Codonopsis Pilosulae Radix* 15 g, *Acontii Lateralis Radix Preparata* 6 g, *Ephedrae Herba* 6 g, *Glycyrrhizae Radix et Rhizoma* 6 g	
Wang (2018) [34]	1 month	Three times a day	*Astragali Radix* 45 g, *Cinnamomi Ramulus* 15 g, *Paeoniae Radix Alba* 20 g, *Zingiberis Rhizoma Recens* 18 g, *Zizyphi Fructus* 4 pieces, *Cocicis Semen* 30 g, *Mucunae Caulis* 30 g, *Salviae Miltiorrhizae Radix* 15 g, *Atractyodis Rhizoma* 15 g, *Achyranthis Radix* 15 g, *Citri Unshius Pericarpium* 12 g, *Angelicae Gigantis Radix* 12 g, *Scorpio* 10 g, *Carthami Flos* 10 g, *Persicae Semen* 10 g, *Atractylodis Rhizoma alba* 10 g, *Saposhnikoviae Radix* 10 g	- Wind–cold at upper extremity: Osterici Radix 10 g, Curcumae Longae Rhizoma 10 g, Gentianae Macrophyllae Radix 15 g - Wind–cold at lower extremity: Araliae Continentalis Radix 10 g, Sinomeni Caulis et Rhizoma 12 g, Piperis Kandsurae Caulis 12 g - Dual deficiency of qi and blood: Lycii Fructus 15 g, Polygoni Multiflori Radix 15 g, Rehmanniae Radix Preparata 10 g - Kidney deficiency: Drynariae Rhizoma 30 g, Loranthi Ramulus, Dipsaci Radix 15 g, Eucommiae Cortex 15 g, Loranthi Ramulus 12 g, Cibotii Rhizoma 12 g - Qi deficiency and spontaneous sweating: Trtici Levis Semen 30 g, Ephedrae Radix Et Rhizoma 12 g - Chronic impediment disease: 2 of Scolopendrae Corpus, Asiasari Radix et Rhizoma 5 g
Li (2016) [35]	8 weeks	Three times a day	*Cinnamomi Ramulus* 10 g, *Poria Sclerotium* 20 g, *Moutan Cortex* 10 g, *Paeoniae Radix Alba* 15 g, *Persicae Semen* 10 g, *Curcumae Longae Rhizoma* 15 g, *Acontii Lateralis Radix Preparata* 10 g, *Asiasari Radix et Rhizoma* 6 g, *Angelicae Gigantis Radix* 10 g, *Zingiberis Rhizoma* 10 g, *Scorpio* 5 g, *Astragali Radix* 20 g, *Glycyrrhizae Radix et Rhizoma* 6 g	
Zhao (2018) [36]	10 weeks	Once a day	*Poria Sclerotium* 20 g, *Cinnamomi Ramulus* 10 g, *Paeoniae Radix Alba* 15 g, *Curcumae Longae Rhizoma* 15 g, *Moutan Cortex* 10 g, *Asiasari Radix et Rhizoma* 6 g, *Zingiberis Rhizoma* 10 g, *Angelicae Gigantis Radix* 10 g, *Astragali Radix* 20 g, *Glycyrrhizae Radix et Rhizoma* 6 g	
Mao (2020) [37]	14 days	Once a day	*Gingseng Radix Rubra* 15 g, *Astragali Radix Preparata* 20 g, *Acontii Lateralis Radix Preparata* 8 g, *Cinnamomi Ramulus* 15 g, *Asiasari Radix et Rhizoma* 3 g, *Osterici Radix* 9 g, *Araliae Continentalis Radix* 12 g, *Eucommiae Cortex* 15 g, *Achyranthis Radix* 15 g, *Liriopis Tuber* 20 g, *Angelicae Gigantis Radix* 15 g, *Cnidii Rhizoma* 10 g, *Mucunae Caulis* 15 g, *Asini Corii Colla* 10 g, *Piperis Kandsurae Caulis* 15 g, *Corydalis Tuber* 15 g, *Clematidis Radix* 20 g, *Paeoniae Radix Alba* 30 g, *Bupleuri Radix* 9 g, *Chaenomelis Fructus* 9 g, *Citri Unshius Pericarpium* 8 g, *Glycyrrhizae Radix et Rhizoma* 6 g	- Static blood: Olibanum, Myrrha, Persicae Semen - Phlegm: Trichosanthis Radix - Difficulty bending and straightening: Lycopodii Herba - Inability to sleep: Zizyphi Spinosi Semen, Polygoni Multiflori Caulis - Reduced food intake: Galli Stomachichum Corium
Zhang (2020) [38]	10 days	Twice a day	*Araliae Continentalis Radix* 9 g, *Loranthi Ramulus* 6 g, *Eucommiae Cortex* 6 g, *Achyranthis Radix* 6 g, *Asiasari Radix et Rhizoma* 6 g, *Gentianae Macrophyllae Radix* 6 g, *Poria Sclerotium* 6 g, *Cinnamomi Cortex* 6 g, *Saposhnikoviae Radix* 6 g, *Cnidii Rhizoma* 6 g, *Gingseng Radix* 6 g, *Glycyrrhizae Radix et Rhizoma* 6 g, *Angelicae Gigantis Radix* 6 g, *Paeoniae Radix Alba* 6 g, *Rehmanniae Radix* 6 g	-
Wu (2019) [39]	7 days	Twice a day	*Loranthi Ramulus* 20 g, *Achyranthis Radix* 20 g, *Rehmanniae Radix Preparata* 20 g, *Eucommiae Cortex* 20 g, *Gentianae Macrophyllae Radix* 15 g, *Araliae Continentalis Radix* 15 g, *Saposhnikoviae Radix* 15 g, *Paeoniae Radix Alba* 15 g, *Angelicae Gigantis Radix* 15 g, *Cnidii Rhizoma* 15 g, *Codonopsis Pilosulae Radix* 15 g, *Poria Sclerotium* 15 g, *Cinnamomi Cortex* 10 g, *Asiasari Radix et Rhizoma* 10 g, *Glycyrrhizae Radix et Rhizoma* 10 g	-

**Table 3 healthcare-11-02743-t003:** Characteristics of conventional treatment interventions in the included studies.

Author, Year	Ingredient	Dose	Administration Route	Frequency	Duration
Lu (2020) [31]	Ibuprofen	1 capsule (0.3 g)	Oral	Twice a day	1 month
Zhang (2019) [32]	Ibuprofen	1 capsule (0.3 g)	Oral	Twice a day	1 month
Chen (2021) [33]	Indomethacin	N.R.	Cataplasm	Once a day	3 months
Wang (2018) [34]	Indomethacin	N.R.	Cataplasm	Once a day	1 month
Li (2016) [35]	Indomethacin	N.R.	Cataplasm	Once a day	8 weeks
Mao (2020) [36]	Indomethacin	N.R.	Cataplasm	Once a day	10 weeks

N.R., Not recorded.

**Table 4 healthcare-11-02743-t004:** Measurement method of TER for puerperal wind syndrome in the included studies.

Author, Year	Scale of TER	Symptoms Included in the TER Evaluation						
		Pain	Discomfort	Coldness	Numbness	Swelling	Functional Activity	Fatigue
Lu (2020) [31]	3-point scale	O	O	X	O	O	O	X
Zhang (2019) [32]	3-point scale	O	X	X	O	X	O	X
Chen (2021) [33]	4-point scale	N.R.	N.R.	N.R.	N.R.	N.R.	N.R.	N.R.
Wang (2018) [34]	4-point scale	O	X	O	O	X	O	O
Li (2016) [35]	4-point scale	O	X	O	O	X	X	O
Mao (2020) [37]	3-point scale	O	X	X	X	X	O	X
Zhang (2020) [38]	4-point scale	N.R.	N.R.	N.R.	N.R.	N.R.	N.R.	N.R.
Wu (2019) [39]	3-point scale	O	X	X	X	X	O	X

N.R., Not recorded.

**Table 5 healthcare-11-02743-t005:** Summary of findings for visual analog score (VAS).

Outcome	Number of Participants (Studies)	Certainty of the Evidence (GRADE)	Relative Effect (95%)	Anticipated Absolute Effects	
				Assumed Risk	
				Placebo	Risk Difference with Pain (VAS)
Herbal medicine versus conventional treatment	217 (3 studies)	⨁⨁⨁◯ Moderate	-	-	MD 1.4 lower (2.02 lower to 0.78 lower)
Herbal medicine plus warm needle acupuncture versus warm needle acupuncture	64 (1 study)	⨁⨁◯◯ Low	-	-	MD 1.4 lower (2.02 lower to 0.78 lower)
Herbal medicine plus moxibustion versus moxibution	60 (1 study)	⨁⨁◯◯ Low	-	-	MD 1.4 lower (2.02 lower to 0.78 lower)
Herbal medicine plus Chuna versus Chuna	58 (1 study)	⨁⨁◯◯ Low	-	-	MD 1.28 lower (1.85 lower to 0.71 lower)

**Table 6 healthcare-11-02743-t006:** Summary of findings for total effective rate.

Outcome	Number of Participants (Studies)	Certainty of the Evidence (GRADE)	Relative Effect (95%)	Anticipated Absolute Effects	
				Assumed Risk	
				Placebo	Risk Difference with TER
Herbal medicine versus conventional treatment	369 (5 studies)	⨁⨁⨁◯ Moderate	RR 1.19 (1.10 to 1.29)	772 per 1000	147 more per 1000 (77 more to 224 more)
Herbal medicine plus warm needle acupuncture versus warm needle acupuncture	64 (1 study)	⨁⨁◯◯ Low	RR 1.21 (0.96 to 1.52)	750 per 1000	157 more per 1000 (30 fewer to 390 more)
Herbal medicine plus moxibustion versus moxibution	60 (1 study)	⨁⨁◯◯ Low	RR 1.13 (0.91 to 1.39)	800 per 1000	104 more per 1000 (72 fewer to 312 more)
Herbal medicine plus Chuna versus Chuna	58 (1 study)	⨁⨁◯◯ Low	RR 1.29 (1.01 to 1.64)	724 per 1000	210 more per 1000 (7 more to 463 more)

**Table 7 healthcare-11-02743-t007:** Summary of findings for scores of Traditional Chinese Medicine.

Outcome	Number of Participants (Studies)	Certainty of the Evidence (GRADE)	Relative Effect (95%)	Anticipated Absolute Effects	
				Assumed Risk	
				Placebo	Risk Difference with Scores of TCM Syndromes
Herbal medicine versus conventional treatment	217 (3 studies)	⨁⨁⨁◯ Moderate	-	-	MD 4.05 lower (6.05 lower to 2.05 lower)

**Table 8 healthcare-11-02743-t008:** Summary of findings for Oswestry Disability Index (ODI).

Outcome	Number of Participants (Studies)	Certainty of the Evidence (GRADE)	Relative Effect (95%)	Anticipated Absolute Effects	
				Assumed Risk	
				Placebo	Risk Difference with ODI
Herbal medicine plus Chuna versus Chuna	58 (1 study)	⨁⨁◯◯ Low	-	-	MD 4.92 lower (7.32 lower to 2.52 lower)

**Table 9 healthcare-11-02743-t009:** Summary of findings for SF-36.

Outcome	Number of Participants (Studies)	Certainty of the Evidence (GRADE)	Relative Effect (95%)	Anticipated Absolute Effects	
				Assumed Risk	
				Placebo	Risk Difference with QOL
SF-36 (Role-emotional)	172 (2 studies)	⨁⨁⨁◯ Moderate	-	-	MD 14.33 higher (13.13 higher to 15.53 higher)
SF-36 (Social functioning)	172 (2 studies)	⨁⨁⨁◯ Moderate	-	-	MD 11.14 higher (2.21 lower to 24.5 higher)

## Data Availability

The data presented in this study are available on request from the corresponding author.

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
