# Peer review of "Herbal Medicine for Postpartum Pain: A Systematic Review of Puerperal Wind Syndrome (Sanhupung)"

_healthcare, 2023, doi:10.3390/healthcare11202743_

Round 1
Reviewer 1 Report
Well designed SR. Few modifications are needed
1. Add participants characteristics of the included studies to table 1
2. Write the exact search strategy for each database in a supplementary table
3. Add causes of exclusion of the excluded studies
4. Details about the registration of the included studies
5. Add quality of evidence to your findings GRADE
6. Adequate subgroup analysis for different forms, timing, doses, and duration of uses of different herbal medicines
Adequate
Reviewer 2 Report
The group of authors tried to perform metanalysis on RCTs in Eastern traditional medicine to alleviate the symptoms of postpartum pain.
1) The topic could be considered original if the authors could define the underlying molecular mechanism of the plants used in the decoction.
2) The most specific improvement is required regarding breastfeeding mothers. If the authors would not be able to demonstrate the safety of the plants used in these RCTs for the infants during breastfeeding with appropriate citations, the manuscript cannot be considered further.
3) The manuscript needs in the introduction section to introduce that each symptom in postpartum pain can originate from which potential etiologies with appropriate references. Could it be derived from oxytocin or other disorders such as diabetes mellitus or hypothyroidism, or cardiac failure during or after pregnancy?
Besides a complete response to the above three comments, some other improvements are required as follows:
4) Genus and plant species should be in italics throughout the manuscript, such as section 3.3.2. The authors should note that the genus name must be initiated with a capital letter but not the species
5) In section 3.4.1, the authors should clearly define the name of drugs, dose, and duration of treatment for what they called “Western medicine treatment”.
6) The caption of Figures 3-8 should completely describe the details of the figure independently.
Reviewer 3 Report
Kwon et al. tried to address two important research questions: the safety of herbal medicines and treatment for postpartum pain. Both are well-reviewed, and this information benefits further research for clinicians and biologists in this important field.
-
The sample size was large enough to support their conclusions.
-
References were very well organized.
-
The tables were good and the statistical analysis was suitable for the interpretation.
However, I suggest some minor changes.
-
Are there any high-throughput methods used in this field of study, like transcriptomics? If yes, please include.
-
A scheme can be included to show your overall findings.
-
I suggest also writing about other traditional herbal medicinal systems, like Ayurveda, Siddha and Unani.
Round 2
Reviewer 2 Report
The manuscript is improved after revision.